# Impact of acute open-skill exercise on inhibitory control and brain activation: A functional near-infrared spectroscopy study

**Shinji Takahashi**[1]*, **Philip M. Grove**[2]

**1** Faculty of Liberal Arts, Tohoku Gakuin University, Sendai, Japan, **2** School of Psychology, The University of Queensland, Brisbane, Australia

* shinji@mail.tohoku-gakuin.ac.jp

**Data Availability Statement:** All relevant data are within the paper and its Supporting information files.

**Funding:** This study is a part of the research project "Influence of different types of acute

## Abstract

There is a growing body of literature demonstrating that a single bout of exercise benefits executive cognitive function. While the acute effect of closed-skill exercises like walking, running, and cycling has been well investigated, it is less clear how open-skill exercise impacts executive function and brain activation. Therefore, we compared the acute effects of an open-skill exercise on inhibitory control and brain activation with those of a closed-skill exercise using functional near-infrared spectroscopy (fNIRS). Twenty-four young right-hand dominant adults (9 women) completed three interventions: badminton, running, and a seated rest control condition for 10 min each. The intensities of badminton and running were comparable. During each intervention, oxygen uptake and heart rate were monitored. A Stroop task composed of neutral and incongruent conditions was administrated before and after each intervention. An fNIRS system recorded hemodynamics in the prefrontal cortex to evaluate brain activation during the Stroop task. Performance on the Stroop task was significantly improved after badminton, specifically in the incongruent condition relative to in the neutral condition. On the other hand, neither running nor seated rest affected performance in the Stroop task. The fNIRS measures indicated that badminton and running had no significant influence on brain activation. These results show that a single bout of open-skill exercise enhances inhibitory control without increasing brain activation compared to closed-skill exercise, suggesting that an acute open-skill exercise induces neural efficiency.

## Introduction

Regular exercise is known to play an essential role in both preventing or slowing cognitive decline in the elderly [1, 2] and enhancing the development of cognitive functions for children and adolescents [3–5] with several studies pointing to an association between exercise and cognition and its underlying mechanism. Investigations on the effect of a single bout of exercise on cognitive functions [6–9] have revealed effects on executive functions that include inhibitory control, working memory, and cognitive flexibility [10, 11]. So far, the accumulated evidence indicates that single bouts of aerobic exercise have a small but positive effect [5–8, 12],

exercises on cognitive functions and brain activation," supported by the Japan Society for the Promotion of Science (Grant number JP 18K10855). The author ST received the grant. The funders had no role in study design, data collection and analysis, decision to publish, or preparation of the manuscript.

**Competing interests:** The authors have declared that no competing interests exist.

modulated by exercise intensity, duration, the timing of measures for executive functions [12], developmental stage [7], and baseline executive functions levels for individuals [6].

Although quantitative exercise variables such as intensity and duration are well studied, qualitative features like exercise type have received less attention. In particular, the impact of acute open-skill exercises, requiring adaptation to dynamic environments and social interaction between teammates and opponents, on executive functions is poorly understood. Regarding regular open-skill exercises, several systematic reviews indicate that regular open-skill exercises (e.g., basketball, football, and tennis) benefit executive functions more than closed-skill sports (e.g., running and swimming) [3, 11, 13, 14]. Voss et al. [13] found that processing speed, a measure of cognitive function, benefits from interceptive sports (e.g., tennis, squash) and strategic sports (e.g., volleyball, basketball, soccer). In contrast, static sports like long-distance running and swimming do not significantly affect processing speed. Contreras-Osorio et al. [3] demonstrated that sports interventions that involve open-skill exercises over several days have significant effects on the overall executive functions of children and teenagers.

Given the benefits of regular open-skill exercises on executive functions, it is expected that open-skill exercise has a different effect on executive functions than closed-skill exercise for even single bouts of exercise. There have been several studies that investigated the various effects of acute exercise types on executive functions; however, the previous results are mixed [15–23].

Pesce et al. [17] reported that a team-based activity involving cognitive demands and social interaction enhanced free-recall word memory performance immediately after the activity. In contrast, a circuit activity with fewer cognitive demands and social interaction failed to show a similar effect. Cooper et al. [23] also demonstrated a significant impact of the team-based activity on inhibitory control and working memory. They found these processes were enhanced following basketball relative to seated rest. Similarly, Takahashi and Grove [18] found that singles-game badminton enhanced inhibitory control compared with running or seated rest. These results support the idea that a single bout of open-skill exercise benefits executive functions relative to closed-skill exercise.

In contrast, other studies have reported that the effect of a single bout of open-skill exercise on executive functions is comparable to or less than that of closed-skill exercises. O'Leary et al. [21] found that although walking enhanced the inhibitory control, exergaming composed of aerobic and cognitive domains did not positively affect the inhibitory control. Gallotta et al. [20] also failed to show a benefit of open-skill exercise on cognitive functions. Gallotta et al. compared basketball and running with seated rest, revealing that running and seated rest enhanced attention and concentration more than basketball. Both O'Leary and Gallotta et al. speculated that the decision-making and the strategic manner required in open-skill exercises were additional cognitive loads, resulting in attenuating benefits of aerobic exercise on executive functions. The speculations are consistent with previous studies [19, 24] showing that adding a cognitive task to a closed-skill exercise condition increases cognitive burden compared with a closed-skill only exercise condition or a cognitive task only condition.

Although different impacts of open-skill exercises have been shown, studies reporting both positive and negative effects imply that a single bout of open-skill exercise would activate the brain more than closed-skill exercises. Visser et al. [25] reported an interesting result regarding brain activation during open-skill exercise. Visser et al. compared nervous activity measured by electroencephalography (EEG) during table tennis with that during cycling, revealing increased nervous activity in the frontal region during table tennis relative to cycling. The authors demonstrated greater activation in the region of the brain underpinning the executive functions (e.g., prefrontal cortex) during an open-skill exercise relative to during a closed-skill exercise. However, to the best of our knowledge, no study has compared the impact of closed-

skill and open-skill exercise on brain activation during an executive function task following an acute bout of exercise. Therefore, we measured brain activation during an inhibitory control task after a single session of closed-skill exercise, open-skill exercise, and a rest control.

We compared behavioral performance (reaction time and accuracy) and hemodynamics in the prefrontal cortex after a bout of badminton, running, or seated rest in a cross-over experimental design, using a color-word Stroop task [26, 27] and functional near-infrared spectroscopy (fNIRS). Badminton was chosen as an open-skill exercise in this study because badminton requires various motions and cognitive demands. Running was the closed-skill exercise in line with previous research. Given previous reports that a single aerobic exercise session improves behavioral measures of the Stroop task with increased hemodynamics in the prefrontal cortex [28–34], we hypothesized that an open-skill exercise like badminton would shorten reaction times in the Stroop task and increase hemodynamics in the prefrontal cortex compared to running.

## Materials and methods

### Participants

We used G*Power version 3.1.9.4 software package (Düsseldorf, Germany) to conduct a power analysis for a one-way repeated analysis of variance (ANOVA) to calculate a sufficient sample size under the following conditions: the dependent variable was the change of the reaction time for the incongruent condition Stroop trial between pre-intervention and post-intervention; the independent variable was exercise mode with three levels: badminton, running, and control intervention; partial eta squared was 0.05 ($f = 0.23$); power ($1 − β$) was 0.95; the intraclass correlation coefficient was 0.8 [35] and alpha at 0.05. This analysis indicated that the sample size of 22 was adequate. Participants were recruited through sports and health sciences courses at Tohoku Gakuin University between December 2019 and July 2020. The recruitment criteria were 1) right-hand dominant undergraduate students, 2) normal or corrected to normal vision, and 3) no history of brain, cognition, mental, or cardiovascular diseases. Twenty-four healthy right-hand dominant Japanese (9 women) voluntarily participated in this study. The participants were asked to refrain from alcohol use and strenuous physical activity for 24 h before each experiment and to refrain from smoking, food, or caffeine consumption for two hours preceding the experiments. Written informed consent was obtained from all participants before the first experiment. The Human Subjects Committee of Tohoku Gakuin University approved the study protocol (Approval number: 2019R003). All records including the participant's personal information were coded to preserve anonymity. Table 1 shows the characteristics of the participants.

Table 1. Characteristics of the participants (mean ± SE).

| Variable | Total ($N = 24$) | Men ($N = 15$) | Women ($N = 9$) |
|---|---|---|---|
| Age (years) | 20.4 ± 0.2 | 20.5 ± 0.1 | 20.2 ± 0.4 |
| Height (cm) | 168.2 ± 1.7 | 171.9 ± 1.8 | 161.9 ± 1.8 |
| Weight (kg) | 61.6 ± 1.8 | 66.1 ± 2.0 | 54.1 ± 1.9 |
| BMI (kg·m$^{-2}$) | 21.7 ± 0.4 | 22.3 ± 0.5 | 20.6 ± 0.5 |
| IPAQ (MVPA-min·week$^{-1}$) | 2719.7 ± 652.2 | 3758.0 ± 937.1 | 989.3 ± 318.2 |
| $\dot{V}O_2$peak (mL·kg$^{-1}$·min$^{-1}$) | 46.9 ± 1.1 | 49.7 ± 1.1 | 42.1 ± 1.5 |
| HRpeak (bpm) | 192.9 ± 1.8 | 191.8 ± 2.5 | 194.8 ± 2.4 |

BMI: body mass index; IPAQ: International Physical Activity Questionnaire; MVPA: moderate-vigorous physical activity; $\dot{V}O_2$peak: the peak of oxygen uptake; HRpeak: the peak of heart rate.

## Procedures

Participants visited the sports physiology laboratory in the gymnasium on four different days (average interval, $6.1 \pm 1.8$ days).

**Day 1.** Participants received a brief introduction to the study and completed informed consent. Their height and weight were measured using a stadiometer and a digital scale. They then answered a questionnaire about medical history and the International Physical Activity Questionnaire (IPAQ) [36, 37], a measurement tool to assess the quantity of physical activity for the recent week. Next, to familiarize participants with the computer-based color-word Stroop task and an fNIRS device (OEG-16; Spectratech Inc., Yokohama, Japan), they completed the Stroop task while wearing the fNIRS device. A graded exercise test was subsequently conducted to determine the peak of oxygen uptake ($\dot{V}O_2$peak) and the peak of heart rate (HRpeak). After the graded exercise test, the participants completed the Stroop task with the fNIRS device again.

**Day 2–4 (experimental sessions).** Laboratory visits 2, 3, and 4 were experimental sessions. To minimize any order or learning effects, the orders of the experimental sessions were randomized. After arrival at the laboratory, the participants rested on a comfortable chair for 10 min. In the experimental sessions, first, participants were asked about their mood by the two-dimensional mood scale (TDMS) [38] for 1 min to evaluate mental fatigue. They then completed the Stroop task with the sensor of fNIRS on their forehead before and after each intervention. Fitting the fNIRS sensor and calibrating this device took approximately 3 min, and the Stroop task took 4.2 min. After the pre-test of the Stroop task, participants removed the fNIRS sensor and were fitted with a portable indirect calorimetry system (MetaMax-3B; Cortex, Leipzig, Germany) which took 1 min. Participants then rested on a chair for 3 min. For the badminton intervention, the participants moved from the laboratory to a badminton court, which took 2 min. For both the running and the control interventions, the participants walked on a treadmill (O2road, Takei Sci. Instruments Co., Niigata, Japan) at 4.2 km·h$^{-1}$ for 2 min, which served as a counterpart to the move from the laboratory to the badminton court. Subsequently, the participants performed each intervention. Based on the protocol of the previous studies [16, 18, 35], the duration of the intervention was set to 10 min. After each intervention, participants returned to the laboratory or walked on the treadmill for 2 min and then they removed the indirect calorimetry system. Subsequently, they rested for 15 min on a chair. At the end of the resting period, the participants' moods were probed by the TDMS again. Finally, they completed a post-test Stroop task and the fNIRS measure again.

In the badminton intervention, each participant played one singles game against an investigator who had experience teaching badminton in physical education courses. The investigator played at a level of proficiency that matched the participant's level and had social interaction with participants by providing tips. During the game, the scores were not recorded, and "victory or defeat" was not determined. In the running intervention, the participants ran on the treadmill. Running speed was set according to each participant's 75%$\dot{V}O_2$peak, which has been previously shown to be the intensity equal to that of the badminton intervention [18, 35]. In the control intervention, the participants were seated on a comfortable chair with their smartphones and were instructed to spend time operating their smartphones as normal. Oxygen uptake ($\dot{V}O_2$), carbon dioxide output ($\dot{V}CO_2$), and HR were monitored throughout each experimental session using the MetaMax-3B. The MetaMax-3B was calibrated before each intervention. The turbine was calibrated with a 3.0-L calibration syringe, and the oxygen and dioxide sensors were calibrated with room air and a calibration gas of known $O_2$ (16%) and $CO_2$ (4%) composition. The $O_2$ and $CO_2$ sensors were calibrated again using room air prior to monitoring. Physiological measures for the last 7 min were averaged, and the rating of perceived exertion (RPE) was evaluated at the end of each intervention.

### Aerobic fitness assessment

Participants performed the graded exercise test on the motor-driven treadmill to volitional exhaustion. The initial speed was set at 7.2 km·h$^{-1}$. The slope of the treadmill was constant at 1.0%. Each speed lasted 2 min and the speed was increased by 1.2 km·h$^{-1}$ until volitional exhaustion. The MetaMax-3B measured $\dot{V}O_2$, $\dot{V}CO_2$, and HR, and the average of the final 30 s was defined as $\dot{V}O_2$peak and HRpeak. RPE was measured at the end of each stage. Volitional exhaustion was reached based on the following criterion: 1) RPE $\geq$ 17, 2) HR $\geq$ 95% of age-predicted HRmax (220 minus age), and 3) a respiratory exchange ratio (RER, $\dot{V}CO_2 \cdot \dot{V}CO_{2^{-1}}$) $\geq$ 1.10.

### Color-word Stroop task

The computer-based color-word Stroop task, composed of a neutral condition block and an incongruent condition block, assessed inhibitory control for each participant (see Fig 1). The beginning of the Stroop task was the baseline period of 30 s, in which a picture of nature (river scenery) was presented on a 15-inch laptop screen. After the baseline period, either the neutral condition block or the incongruent condition block was started, determined randomly. Each block had 24 trials. For the trials in the neutral condition, the laptop screen presented two rows on a gray background, the upper row was 'XXXX' printed in one of the red, blue, yellow, and green, and the lower row was one of the words 'Red,' 'Blue,' 'Yellow,' and 'Green' printed in white. For the trials in the incongruent condition, the upper row presented either word 'Red,' 'Blue,' 'Yellow,' and 'Green' printed in incongruent colors (e.g., 'Red' was printed in blue), and the lower row was one of the words 'Red,' 'Blue,' 'Yellow,' and 'Green' printed in white. The trials were shown on the screen for 3 s. The upper row was presented 500 ms before the lower row; the response period was 2.5 s. Each trial for the neutral and incongruent conditions was separated by intervals that showed a white cross for 1 s.

The participants were instructed to press the right cursor button with the right hand when the color of the upper row matched the color name in the lower row and to press the left cursor

**Fig 1. Scheme of the color-word Stroop task.** The scheme consists of two blocks the neutral condition block and the incongruent condition block. The order of the two blocks was randomized.

button with the right hand when the color of the upper row was mismatched with the color name in the lower row. The ratio of the correct responses of the right (matched) and left (mismatched) buttons was 50%, and the order of the two answers was randomly assigned. The participants were asked to respond as quickly and accurately as possible. All words in the task were written in Japanese. Reaction time and the number of errors were recorded.

## Functional near-infrared spectroscopy

The multi-channel fNIRS device (OEG-16) measured hemodynamics in the prefrontal cortex during the Stroop task. This device has a headband module with six emitter probes and six detector probes arranged on a two × six matrix to detect signals from 16 channels. The six emitter probes emit two-wavelength near-infrared light (approximately 770 and 840 nm). The six detector probes placed at 30 mm from the emitter probes detect the re-emitted light. The center of the probe matrix is placed on Fpz (International 10–20 system), and the matrix covers from F7 to F8 (see Fig 2). This device records changes in oxy-hemoglobin (oxy-Hb) and deoxy-hemoglobin (deoxy-Hb) at approximately 30 mm below the scalp. The sampling

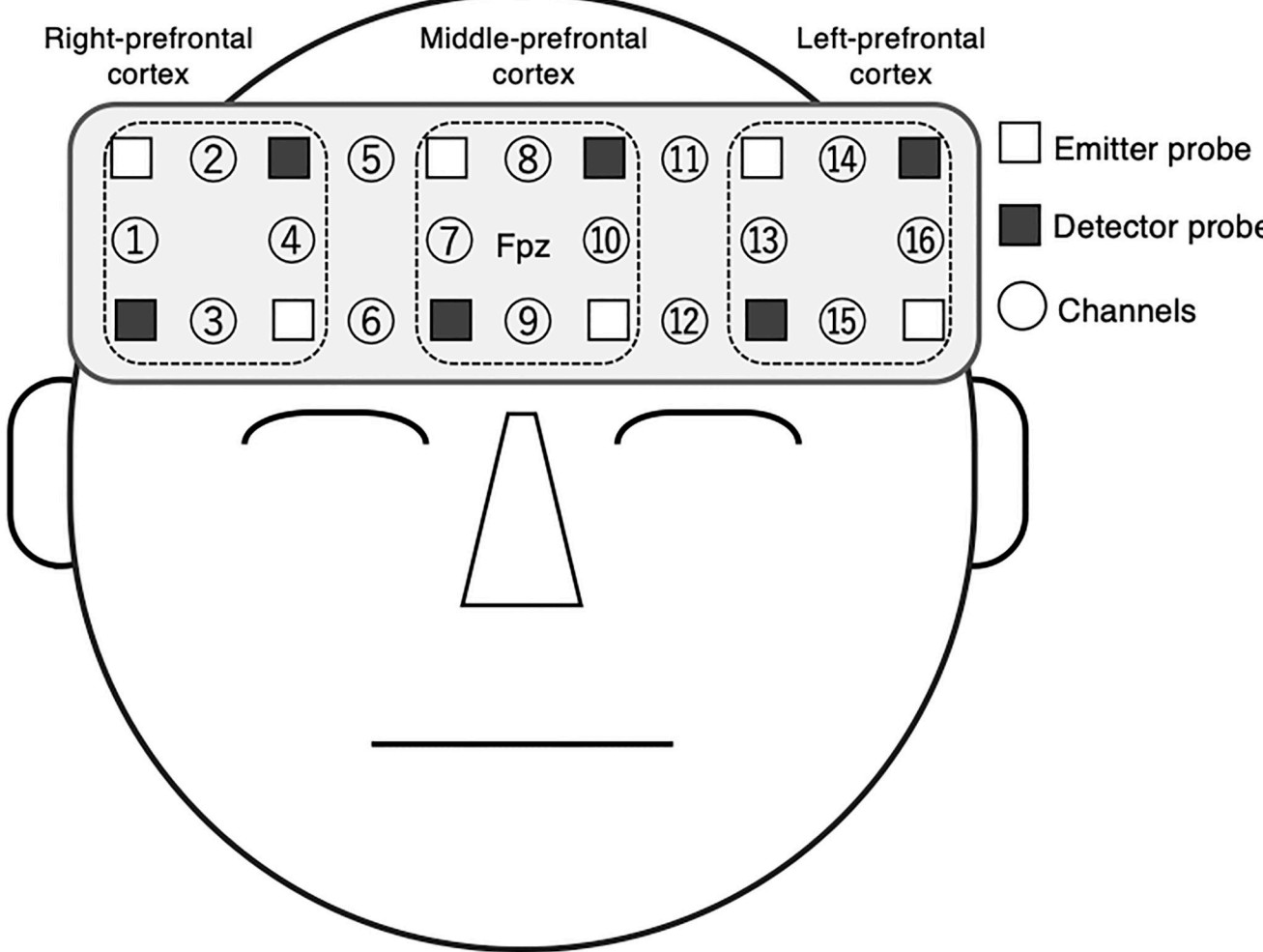

**Fig 2. The layout for emitter and detector probes in the headband module of the NIRS device.** The center of the probe matrix was placed on the Fpz (International 10–20 System).

interval is 0.65 s. We cleaned the emitter, detector probes, and the forehead area for participants with alcohol cotton before each Stroop task.

## Processing the fNIRS data

Raw records of oxy-Hb and deoxy-Hb were band-pass filtered on the specific software for this device (OEG-16 control program). To remove artifacts from drift, respiration, and heartbeat, the high-pass and low-pass filters were set at 0.01 Hz and 0.1 Hz, respectively [39]. Moreover, systemic physiological artifacts were removed using the hemodynamic modality separation method [40]. After this data processing, the raw dataset was exported to a CSV format file. Then, in order to increase the ratio of signal, oxy-HB and deoxy-Hb for each channel were converted to $z$-scores using the mean and standard deviation for the first 13 s (20 samples) of the first baseline in the Stroop task. We used data for the first 13 s of the first baseline for the calculation of $z$-scores because the participants appeared to mentally prepare for the Stroop task so that there was an increase and decrease trend in oxy-Hb and deoxy-Hb during the latter part of the baseline. Since the $z$-scores for oxy-Hb and deoxy-Hb modified by the hemodynamic modality separation method [40] resulted in an identical value, although opposite in sign, we used the oxy-Hb $z$-scores for subsequent analyses.

We averaged the oxy-Hb in every channel for the neutral and incongruent condition blocks. Then, the oxy-Hb $z$-scores for the neutral and incongruent condition blocks were averaged for 1–4 channels (Right-prefrontal cortex), 7–10 channels (Middle-prefrontal cortex), and 13–16 channels (Left-prefrontal cortex), respectively.

## Psychological mood

To assess the subjective mental fatigue of participants that potentially affects behavioral performance for the Stroop task, the TDMS [38] measured psychological pleasure and arousal states before the Stroop tasks. The TDMS was composed of eight items (6 Likert scales) that probed the psychological conditions of energetic, lively, lethargic, listless, relaxed, calm, irritated, and nervous. The range of pleasure and arousal states scored by the TDMS was -20 to 20.

## Statistical analysis

All measurements are described as mean ± standard error of the mean. Statistical analyses were conducted using IBM SPSS 27 (SPSS Inc., Chicago, IL, USA). To examine the exercise intensity of each intervention, %HRpeak, %$\dot{V}O_2$peak, RER, and RPE were compared using a mixed model with Mode (running, badminton, and control) as a fixed effect and participant as a random effect. A significant main effect of Mode was followed up with the Bonferroni corrected $t$-tests.

Psychological moods scored by the TDMS were analyzed using a mixed model with Time (pre-and post-test) and Mode (running, badminton, and control) and the interaction (Time × Mode) as fixed effects and participant as a random effect for pleasure level and arousal level, respectively. If there was a significant interaction, psychological moods were compared using the mixed model with Mode as a fixed effect and participant as a random effect, and the Bonferroni corrected $t$-tests for pre-test and post-test, respectively.

The reaction time for the Stroop tasks was compared using a mixed model with Condition (neutral and incongruent), Time (pre-and post-test), and Mode (running, badminton, and control), interactions (Condition × Time, Condition × Mode, Time × Mode, and Condition × Time × Mode) as fixed effects and participant as a random effect. When any significant interactions were found, the mixed models with two fixed effects composing the significant interactions and the participant as a random effect were examined in a post hoc

analysis. Also, the number of errors was compared using a log-linear mixed model with Condition (neutral and incongruent), Time (pre-and post-test), and Mode (running, badminton, and control), interactions (Condition × Time, Condition × Mode, Time × Mode, and Condition × Time × Mode) as fixed effects and participant as a random effect.

Similarly, the oxy-Hb for fNIRS was analyzed using a mixed model with Condition, Time, Mode, and interactions as fixed effects and participant as a random effect, for the Right-, Middle-, and Left-prefrontal cortex, respectively.

We set the covariance structure as the unstructured for all the mixed models. Significance levels for all analyses were set at $P = .05$. Cohen's $d$ was calculated to assess the effect sizes for differences between the two means.

## Results

To ensure whether the order of interventions and the order of the Stroop task condition affected the results, we conducted analyses with the mixed models adding the orders of the interventions and the Stroop task condition as preliminary analyses. The preliminary analyses revealed no impact of the orders of the interventions and the conditions on the results.

### Intensity of interventions

Table 2 presents the intensities for each intervention. The mixed models for $\%\dot{V}O_2peak$, %HRpeak, RER, and RPE revealed significant main effects ($Fs (2, 23) \geq 15.4$, $Ps < .001$). The %$\dot{V}O_2peak$, %HRpeak, RER, and RPE during both the badminton and running interventions were significantly higher than during the control intervention ($Ps < .001$, Cohen's $ds \geq 1.14$). Differences in all intensity measures between the badminton and running interventions were not significant ($Ps \geq .084$, Cohen's $ds \leq |0.50|$).

### Psychological mood

Table 3 shows the changes in the pleasure and arousal states scored by the TDMS for each intervention. The mixed model for the pleasure state did not reveal significant main effects of Mode ($F (2, 23) = 2.9$, $P = .072$), Time ($F (2, 23) = 0.5$, $P = .475$) or the interaction of Mode × Time ($F (2, 23) = 2.4$, $P = .112$). The mixed model for the arousal state revealed a

**Table 2. Intensities of each intervention (mean ± SE).**

| Variable | Intervention | Mean± SE ($N = 24$) |
|---|---|---|
| $\%\dot{V}O_2peak$ (%) | Badminton | 77.9 ± 1.9 * |
| | Running | 76.9 ± 0.9* |
| | Control | 9.2 ± 0.3 |
| %HRpeak (%) | Badminton | 83.3 ± 1.5 * |
| | Running | 87.3 ± 0.9 * |
| | Control | 38.0 ± 1.1 |
| RER ($\dot{V}CO_2 \cdot \dot{V}CO_{2^{-1}}$) | Badminton | 0.96 ± 0.02 * |
| | Running | 0.98 ± 0.01 * |
| | Control | 0.89 ± 0.01 |
| RPE | Badminton | 12.6 ± 0.4 * |
| | Running | 13.6 ± 0.3 * |
| | Control | 6.2 ± 0.3 |

* Significantly different from control intervention; $p < .05$ at Bonferroni multiple comparison tests.

**Table 3. Psychological states in each intervention (mean ± SE).**

| Variable | Intervention | Pre-test | Post-test |
|---|---|---|---|
| Pleasure state (arbitrary unit) | Badminton | 8.7 ± 0.9 | 10.0 ± 0.8 |
| | Running | 8.3 ± 1.0 | 8.3 ± 0.8 |
| | Control | 8.2 ± 0.8 | 8.3 ± 1.0 |
| Arousal state (arbitrary unit) | Badminton | -3.5 ± 0.7 | 2.7 ± 0.7* |
| | Running | -3.7 ± 0.8 | 1.4 ± 1.0* |
| | Control | -4.6 ± 0.6 | -4.9 ± 0.9 |

* Significantly different from control intervention for post-test; $p < .05$ at Bonferroni multiple comparison tests.

significant interaction of Mode × Time ($F(2, 23) = 14.4$, $P < .001$) and the main effects of Mode ($F(2, 23) = 20.5$, $P < .001$) and Time $F(1, 23) = 34.3$, $P < .001$). To decompose the significant interaction of Mode × Time for the arousal state, we compared the arousal state of each intervention for pre-test and post-test, respectively. While there was no significant main effect of Mode ($F(2, 23) = 1.8$, $P = .326$) for pre-test, the main effect of Mode was significant ($F(2, 23) = 23.4$, $P < .001$) for post-test. The arousal states after the badminton and running interventions were significantly higher than after the control intervention ($Ps < 0.001$, Cohen's $d$s ≥ 1.01). The difference between the badminton and the running interventions at post-test was not significant ($P = .981$, Cohen's $d = 0.23$).

## Color-word stroop task

The log-linear mixed model for the number of errors of the color-word Stroop task revealed a significant main effect of Condition ($F(1, 23) = 48.6$, $P < 0.001$) but none of the other interactions or main effects were significant ($Fs(1–2, 23) ≤ 1.9$, $Ps ≥ .174$). The number of errors for the neutral condition was significantly lower than for the incongruent condition ($0.2 ± 0.1$ counts vs. $1.2 ± 0.1$ counts, $P < .001$).

Fig 3 presents the changes in the reaction time for each Mode. The mixed model for the reaction time revealed a significant interaction of Mode × Time ($F(2, 23) = 5.6$, $P = .011$), a main effect of Time ($F(1, 23) = 14.6$, $P < .001$), and a main effect of Condition ($F(1, 23) = 29.9$, $P < .001$). The main effect of Mode ($F(2, 23) = 0.6$, $P = .572$) and the other interactions ($Fs(1–2, 23) ≤ 1.5$, $Ps ≥ .243$) were not significant. To decompose the significant interaction of Mode × Time, we analyzed reaction time using the mixed model with Time, Condition, and Time × Condition as the fixed effects and the random effect of the participants for each intervention.

For the badminton intervention, the interaction of Time × Condition ($F(1, 23) = 7.3$, $P = .013$), the main effects of Time ($F(1, 23) = 26.7$, $P < .001$), and Condition ($F(1, 23) = 16.9$, $P < .001$) were all significant. The significant interaction showed that reaction times were shorter for badminton for the incongruent condition (Cohen's $d = -0.71$) relative to that for the neutral condition (Cohen's $d = -0.48$). For the running and control interventions, although the main effects of Condition were significant ($Fs(1, 23) ≥ 16.9$, $Ps < .001$), the interactions of Condition × Time ($Fs(1,23) ≤ 0.1$, $Ps ≥ .734$) and the main effects of Time ($Fs(1, 23) ≤ 1.3$, $Ps ≥ .263$) were not significant.

## Hemodynamics in the prefrontal cortex

Fig 4 presents the changes in oxy-Hb ($z$-score) during the Stroop task from the pre-test to post-test for the right-, middle-, and left-prefrontal cortex, respectively. Mixed model analysis

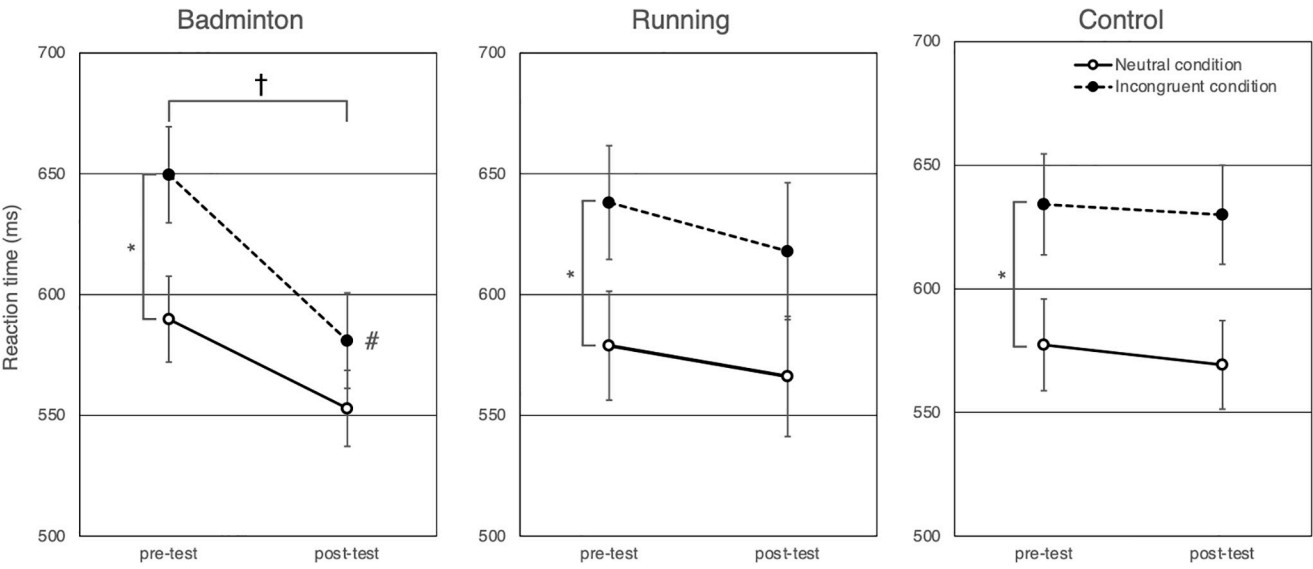

**Fig 3. Comparisons of the reaction time for each intervention.** Solid lines with white circles represent changes in the reaction time for the neutral condition. Broken lines with black circles represent changes in the reaction time for the incongruent condition. Error bars show the standard errors of the mean. For the badminton intervention, the reaction time for both neutral and incongruent conditions was shortened, and the decrease in the reaction time for the incongruent condition was more pronounced than for the neutral condition. Changes between pre-test and post-test were not significant for running and control interventions. * means the significant difference between the neutral condition and the incongruent condition ($p < .05$), † means the significant difference between pre-test and post-test ($p < .05$), and # means that the reaction time for the incongruent condition was significantly shortened than that for the neutral condition ($p < .05$).

did not reveal any significant main effects or interactions for the right-prefrontal cortex region ($Fs (1–2, 23) \leq 2.8$, $Ps \geq .079$). There were no significant main effects or interactions for the middle-prefrontal cortex region ($Fs (1–2, 23) \leq 3.0$, $Ps \geq .096$). A significant main effect of Condition ($F (1, 23) = 9.8$, $P = .005$) for the left-prefrontal cortex was found but no other main effects or interactions were significant ($Fs (1–2, 23) \leq 2.5$, $Ps \geq .104$). These results indicated that oxy-Hb levels in the left-prefrontal cortex significantly increased in the incongruent condition relative to the neutral condition (the incongruent condition $3.3 \pm 0.3$ vs. the neutral condition $2.8 \pm 0.3$, Cohen's $d = 0.55$), regardless of the timing relative to exercises or rest.

## Discussion

This study investigated the impact of a single bout of open-skill exercise on inhibitory control and brain activation compared to closed-skill exercise. To the best of our knowledge, this study is the first to measure brain activation after a single bout of open-skill exercise. The main findings of this study were that a single bout of badminton yielded shorter reaction times for the neutral and incongruent conditions of the Stroop task compared to closed-skill exercise and a rest control while maintaining the same accuracy response. Furthermore, the acute effect of badminton on performance in the incongruent Stroop condition, requiring a greater amount of inhibitory control [41], was more prominent than in the neutral condition. On the other hand, no differences between badminton and running and between pre-test and post-test were observed in the oxy-Hb for each prefrontal cortex region. Taken together, our results indicate that a single bout of open-skill exercise enhances inhibitory control compared to closed-skill exercise without increased hemodynamics in the prefrontal cortex underpinning the inhibitory control. This suggests that an acute open-skill exercise leads to more efficient functioning of the prefrontal cortex, enhancing executive functions relative to acute closed-skill exercise.

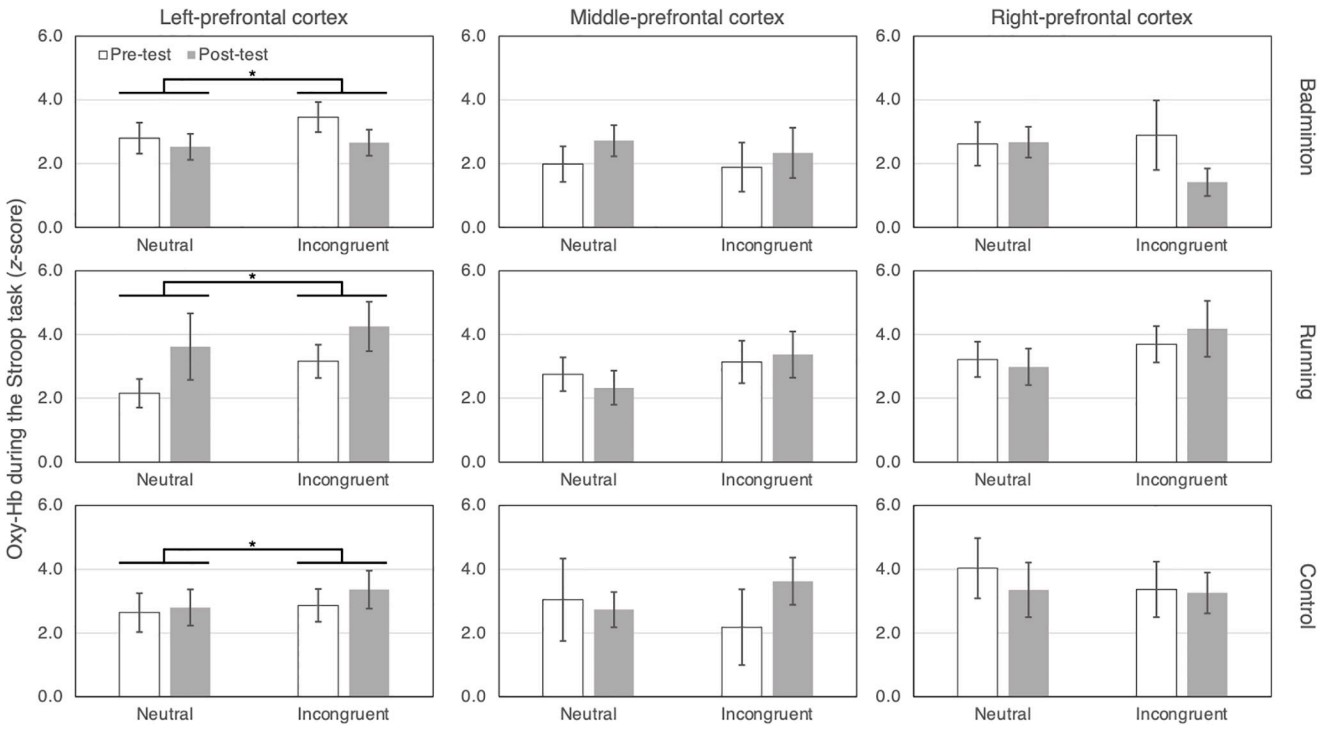

**Fig 4. Comparisons of the oxygenated hemoglobin (z-score) for each intervention and region of the prefrontal cortex.** White bars represent oxygenated hemoglobin at pre-test. Gray bars represent oxygenated hemoglobin at post-test. Error bars show standard errors of the mean. * means the significant difference between the neutral and incongruent conditions ($p < .05$) for the left-prefrontal cortex throughout all interventions.

Both the badminton and running interventions were vigorous-intensity [42] exercises for a short duration of 10 min. The badminton intervention enhanced inhibitory control but this was not found after the running intervention. Our failure to observe enhanced inhibitory control following the running intervention with vigorous-intensity and duration of 10 min is consistent with previous studies. A previous meta-analysis [12] reported that a single session of aerobic exercises with vigorous-intensity has significantly smaller effects on cognitive functions relative to very-light, light, and moderate-intensity exercises. Further, Chang et al. [43] investigated the dose-response relationship between the duration of acute exercise and its effect on inhibitory control, revealing that the exercise time of 10 m was too short to increase performance of inhibitory control. It seems reasonable that a closed-skill with vigorous-intensity and short duration has no effect on inhibitory control.

Although the physiological parameters and exercise-induced changes in psychological moods were comparable during the badminton and running interventions, shorter reaction times for the neutral and incongruent Stroop conditions, maintaining response accuracy, were only observed in the badminton intervention. Additionally, the effect of badminton on the incongruent condition was more pronounced than on the neutral condition. These results suggest that the features of badminton, such as decision-making, various motions, and social interaction, underlie the more extensive effects on inhibitory control, supporting our hypothesis.

Unlike the Stroop task performance results, the fNIRS results were inconsistent with our hypothesis. The mixed model result that the left prefrontal cortex was significantly activated during the incongruent condition relative to the neutral condition matched the previous results [30, 44], indicating the reliability of the fNIRS measure in this study. However, we did

not find any impact of the exercise interventions on hemodynamics. Previous fNIRS studies [28–32] have indicated that the increased hemodynamics in the prefrontal cortex after a closed-skill exercise can improve Stroop task performances. Given the previous results that enhanced inhibitory control results from the exercise-induced activated prefrontal cortex, we hypothesized that an acute open-skill exercise increases oxy-Hb during the Stroop task in the prefrontal cortex relative to a closed-skill exercise, the increasing oxy-Hb improves inhibitory control. However, the shorter reaction times observed in the incongruent condition after badminton did not coincide with increasing oxy-Hb in all prefrontal cortex regions. Prefrontal cortex activation was comparable for both exercise interventions, but improvement in the Stroop task performance was only observed in the badminton intervention. These results suggest that a single bout of open-skill exercises could induce suitable conditions in the prefrontal cortex to achieve goals of inhibitory control tasks, resulting in a decreased Oxy-Hb level required for higher behavioral performance. That is, an acute open-skill exercise could induce neural efficiency [45–47] for executive functions.

Regarding neural efficiency following an acute vigorous-intensity and short-duration exercise, noteworthy findings were reported by Kao et al. [48]. Kao et al. compared 9 min of high-intensity interval training (HIIT) with 20 min of continuous aerobic exercise and 20 min of seated rest, finding that HIIT improved behavioral performances in a flanker task that assesses inhibitory control in similar to a Stroop task, although HIIT decreased the amplitude of P3, indicating lower attentional resource allocation relative to seated rest and continuous aerobic exercise. Further, the latency of P3 reflecting cognitive processing speed was significantly shortened after HIIT compared with seated rest. Still, no difference in the latency of P3 was observed between continuous aerobic exercise and seated rest. Kao et al.'s results indicate that there are different mechanisms in the acute effect of exercise on executive functions, and types of exercise modulate it.

Kao et al. speculated that exercise-induced increasing serum lactate could be a potential reason for neural efficiency after exercise. A vigorous-intensity exercise results in the accumulation of serum lactate, allowing the brain to uptake lactate as an energy source, in addition to glucose. Therefore, they proposed that such a shift in energy sources could induce neural efficiency. However, our results do not seem to support the lactate hypothesis. In this study, because RER ($\dot{V}CO_2 \cdot \dot{V}CO_{2^{-1}}$) correlated with lactate [49, 50] was not different for badminton and running as well as other exercise-intensity indices, serum lactate concentration during badminton seemed to be comparable with during running. Assuming potentially comparable lactate for badminton and running, neural efficiency after badminton might be induced by the features of open-skill exercises.

We speculate that an activated brain during an acute open-skill exercise could be a potential reason for neural efficiency after acute open-skill exercise. Visser et al. [25] found that nervous activities in the brain's frontal region during table tennis measured by EEG were higher than during cycling. Given the significant association between the nervous activities by EEG and the hemodynamics by fNIRS [51], the nervous activities and hemodynamics in the prefrontal cortex during badminton could also be higher than during running. Increased hemodynamics during open-skill exercise could put the brain into a state to exert inhibitory control following cessation of exercise, requiring less brain activation to complete inhibitory control tasks. Any of the features of open-skill exercises such as decision-making, various motions, and social interaction or those interactions could facilitate neural efficiency. Future studies should investigate what cognitive features of open-skill exercises induce neural efficiency.

There are several limitations to this study. First, fNIRS measures are easily affected by different experimental protocols. Previous results [28–32] found a significant association between

enhanced inhibitory control and increased fNIRS measures by an event-related design for cognitive tasks, not a block design which we employed in this study. Compared with event-related design, a block design has the advantage of easily detecting the changes in hemodynamics to cognitive tasks. The disadvantage of a block design is readily contaminated by learning or fatigue effects. Furthermore, the type of experimental design might affect the occurrence of neural efficiency. When the difficulty of a cognitive task is low to moderate, neural efficiency in which behavioral performance is improved without increased brain activation trends occurs [45]. However, a challenging cognitive task increases brain activation and results in a correlation between brain activation and cognitive performance [45]. Thus, if the difficulty of cognitive tasks is also affected depending on the types of experimental design, the neural efficiency might not occur when employing an event-related design that makes cognitive tasks challenging. Previous fNIRS studies [52, 53] that employed block design have also reported neural efficiency. The difficulty of the Stroop task in this study was potentially easier than in the previous studies [28, 44]. The overall difference in the reaction time between the neutral and incongruent conditions in this study was moderate (Cohen's $d$ = 0.62). On the other hand, the previous studies [28, 44] that employed event-related design reported larger effects (Cohen's $d$s $\geq$ 0.80). The gap between previous studies and this study supports that the difficulty of the Stroop task in this study could be low compared with the literature. The experimental design differences may affect the association between cognitive performance and brain activation. Second, the badminton intervention was not a real match. There was no psychological pressure and stress that badminton players must confront in an actual match. In a real badminton match, psychological pressure and stress may influence inhibitory control and brain activation. Third, given the exercise intensities of the badminton and running interventions, HR possibly did not return to their baseline after the two exercise interventions. Using the hemodynamic modality separation method, we removed the systemic physiological artifact, including HR. So, even if HR differed between the exercise and seated rest interventions, we thought the potential effect of differences in HR to fNIRS measures could be removed. In future studies, it is preferred to ensure HR after an exercise returns to its baseline level.

## Conclusions

In conclusion, a single bout of open-skill exercise selectively enhances inhibitory control relative to a closed-skill exercise. On the other hand, hemodynamics in the prefrontal cortex did not differ between the open- and closed-exercise conditions. An acute open-skill exercise may induce neural efficiency after the cession of exercise, enhancing inhibitory control.

## Supporting information

**S1 Data.**
(XLSX)

## Acknowledgments

The authors are grateful to all participants and the badminton instructor for their contributions to this project.

## Author Contributions

**Conceptualization:** Shinji Takahashi, Philip M. Grove.

**Data curation:** Shinji Takahashi.

**Formal analysis:** Shinji Takahashi.

**Funding acquisition:** Shinji Takahashi.

**Investigation:** Shinji Takahashi.

**Methodology:** Shinji Takahashi.

**Project administration:** Shinji Takahashi.

**Resources:** Shinji Takahashi.

**Software:** Shinji Takahashi.

**Supervision:** Philip M. Grove.

**Validation:** Shinji Takahashi.

**Visualization:** Shinji Takahashi.

**Writing – original draft:** Shinji Takahashi, Philip M. Grove.

**Writing – review & editing:** Philip M. Grove.

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
