## [Decision Letter · Decision Letter 0]

25 Nov 2022

PONE-D-22-26975Impact of acute complex exercise on inhibitory control and brain activation: a functional near-infrared spectroscopy studyPLOS ONE

Dear Dr. Takahashi,

Thank you for submitting your manuscript to PLOS ONE. We have now received two reviews by academic specialists in this research field and they have expressed several concerns regarding the current quality of your manuscript. Both reviewers found merit in your work and have addressed important points for improving the manuscript. Please, respond to the comments of the reviewers point by point in a rebuttal letter and change your manuscript based on the guidance of the reviewers.  We feel that your manuscript has merit but does not fully meet PLOS ONE’s publication criteria considering the concerns expressed by the reviewers. Therefore, we invite you to submit a revised version of the manuscript that addresses the points raised during the review process.

We look forward to receiving your revised manuscript.

Kind regards,

Hans-Peter Kubis, PD. Dr. rer. nat.

Academic Editor

PLOS ONE

Journal Requirements:

Reviewers' comments:

Reviewer's Responses to Questions

**Comments to the Author**

1. Is the manuscript technically sound, and do the data support the conclusions?

Reviewer #1: Yes

Reviewer #2: Yes

2. Has the statistical analysis been performed appropriately and rigorously? 

Reviewer #1: Yes

Reviewer #2: Yes

3. Have the authors made all data underlying the findings in their manuscript fully available?

Reviewer #1: Yes

Reviewer #2: Yes

4. Is the manuscript presented in an intelligible fashion and written in standard English?

Reviewer #1: No

Reviewer #2: Yes

5. Review Comments to the Author

Reviewer #1: Impact of acute complex exercise on inhibitory control and brain activation: a functional

near-infrared spectroscopy study

A single bout of exercise is known to be beneficial for execute function. Among exercise already studied, we can find whole-body exercises like running, cycling and walking.

Authors proposed here to compare what they termed “complex” exercise as “simple” exercise on one executive function (inhibitory control with a stroop task) by using functional near-infrared spectroscopy (fNIRS) technique. In other words, they compared the badminton (complex) and running (simple) tasks by carefully matching exercise intensities. The results of this study showed an improved performance on the stroop task for the Badminton task, but without significance influence on brain activation (over the prefrontal cortex) and no significant differences as compared to both running and a control (seated) condition.

General comments

This study presents interesting results on a research topic associating the effects of physical exercise on cognitive functions. Overall, the article appears appropriate, but there are several issues to cope with. Some improvements are required to simplify the reading of the manuscript. Corrections are requested as well as a proofreading by a native person. All sections require to be reduced in length. The introduction must be revised in depth to include a more focused scientific rationale. Methodological aspects of the fNIRS processing require clarification. The discussion tends to be too speculative at times.

Below are the main points to consider.

Introduction

L56. Complex term is likely not appropriate in the abstract and other places of the manuscript.

Motor skill execution is a fundamental part of sporting expertise and a number of recent studies have begun to examine the benefits of visuomotor skills, as those encountered in Badminton on cognitive function. Authors are encouraged to categorize differently exercises they studied.

L62. Static is not the good term to use.

L64-72 and more (e.g. L 84-90, ..). Running cannot be considered simple, and badminton complex. Open skill versus closed skill sports should be used throughout the manuscript according to the scientific background in motor control. Please change accordingly.

Overall, introduction section is rather long. In the rationale, some parts could be removed; for instance, all sentences dealing with mental / cognitive demands and so mental fatigue are out of scope. Stay focus on the executive function of interest (inhibition) that can be modulated in badminton (not really described). Authors are presenting results of literature in executive function as a whole and with different physical tasks but they not addressed basically inhibition in open skill sports like Badminton or Tennis table.

Brain activation is cited but what does it mean? Activation may be due to inhibition or facilitation…Some EEG studies are proposed while authors measured brain activity with other neuroimaging technique (fNIRS) that cannot give relevant information on the inhibitory processing (as the P3 component can do).

L130-132. At the end, hypothesis is not based on a strong rationale

Please rephrase / clarify and simplify the introduction section.

Methods

L143. how did authors assess ‘22’ as an adequate sample size? Please add details on the software used for conducting a Power analysis and the inputs variables (based on which references / findings? It is not reported).

L190. How did authors match and control the workload (volume x intensity) between running and badminton? This is unclear. Table 2 indicates only intensities of each intervention. Ten minutes were used for all interventions?

L204. Can you confirm that Oxygen uptake, carbon dioxide output and HR were monitored during the Badminton session? A portable device was used for that purpose. Please indicate the calibration phases for using the MetaMAx system.

L268-282. Overall, fNIRS processing does not appear to follow current recommendations.

As recommended in the fNIRS field, HbO traces should be analysed concomitantly with HbR signals to make sure that neurovascular coupling response occurred; false positives are possible, especially by looking only at HbO.

In general, the data processing of the fNIRS data is well described. However, I miss a description of how the authors have dealt with systemic physiological artifacts that are known to confound the fNIRS signal (Scholkmann, Tachtsidis, Wolf, & Wolf, 2022; Tachtsidis & Scholkmann, 2016; Yücel et al., 2021).

Can you confirm that the authors use a lowpass filter around 0.09 Hz to account for the Mayer-Waves as recommended in the literature? (Pinti, Scholkmann, Hamilton, Burgess, & Tachtsidis, 2019).

L296. Please add all information (checking) regarding the use of parametric test (one-way repeated ANOVA for physiological variables; two-way repeated ANOVA for psychological moods scores, three-way for the Stroop tasks, fNIRS variables etc.): normality issue, homogeneity of variance. Effect size (partial eta squared) for the main factors should be indicated thereafter in the results.

Did you test some interaction factors according to your hypothesis. Please elaborate.

Results

Individual data should be presented in figures.

Discussion

L447. Intensity (Table 2: 77-78% peak VO2, 84-86% HR, RER close to 0.97 and RPE around 13-14) is claimed vigorous by authors. It is quite surprising to observe a decoupling between RPE values and physiological variables like HR. Please comment.

Can you propose dedicated reference for this statement on vigorous? Ref 41 is not up to date.

According to Jones and Poole, domains of exercise are low/moderate/heavy and severe.

L461-463. Please develop these first ideas for the main significant result you observed.

L472. Authors did not propose some hypothesis on such a correlation; no correlation results were reported.

L498. Since blood lactate samples were not measured, authors cannot state that. Please remove this speculation, out of scope.

From L 500. This section is quite difficult to follow. Please simplify it. When authors are dealing with brain activation, please be accurate on the variables/methods/paradigms used.

L515. There are other limitations related to fNIRS processing (see before). Event-related design with fNIRS is not appropriate as compared to EEG for unveiling underlying mechanisms on the brain changes related to the inhibitory control.

Minor

Reference format: insert a space before the bracket for reference lines 44, 48, 50 and so on.

L195 simple without s

Reviewer #2: This article describes a a study where inhibitory control and brain activity was measured by the Stroop task before and after three interventions: complex exercise (badminton), simple exercise (running), and rest. The study interventions were well-controlled as complex and simple exercises were matched for exertion levels. The authors report that reaction time during the Stroop task was reduced for the badminton interventions but not the other interventions. The authors found no brain activity differences between the conditions, so they interpret their data as showing increased neural efficiency to support the improved performance after badminton. Overall the study design makes sense and the results are not overstated. The authors could provide more context on how their basic results (Stroop performance and brain activity) quantitatively compare to the existing literature.

Minor revisions

1. The sentence starting with “Despite contrary evidence” line 65 page 4 was very confusing and seemed out of place. It’s not clear whether the authors of this paper disagree with Diamond and Ling’s conclusions, or if they are referencing the controversies mentioned in Diamond and Lin’s paper.

2. Figure 2: the circle that is meant to show the head seems to be missing some landmarks like a nose, ears, etc. As it is now it is not very informative and it is hard to understand how low on the forehead the optodes were placed. There also appear to be typos in the legend, should be Emitter Probe and Detector Probe (or just Emitter and Detector).

3. Was it possible to determine if participant’s heart rates had returned to their baseline before the Stroop task was started in the post intervention run?

4. The top bandpass parameter, 0.05 Hz, is quite low. To avoid respiration artifact being below 0.3 Hz should be sufficient. Given the bandpass parameters it would be good to be clear about how long the condition blocks are. It appears that they are at least 3 sec x 24 trials but it’s unclear how long the response period is. Was there only one block of each condition? For future studies you may consider a more classic “block design” with multiple blocks of 16-30 sec in duration.

5. Were any motion artifacts observed in the fNIRS data, and if so how were they handled?

6. There are many explanatory variables in the model (Condition, time, mode, interactions, order of interventions, order of condition) for a comparatively small number of participants (24). The authors should justify that their data supports this complex of a model or consider removing variables.

7. Did the group performance (accuracy, reaction time) on the Stroop task match what was expected based on the literature?

8. Did the difference in brain activation between Stroop conditions for left prefrontal cortex only match existing literature?

9. For Figure 4, I’d suggest using a different color scheme or even grayscale as blue and red are traditionally used to represent oxy- and deoxyhemoglobin. You could also consider putting the Left prefrontal cortex graphs on the left of the figure, and the right prefrontal graphs on the right.

6. PLOS authors have the option to publish the peer review history of their article (what does this mean?). If published, this will include your full peer review and any attached files.

Reviewer #1: No

Reviewer #2: No

---

## [Author Response · Author response to Decision Letter 0]

16 Jan 2023

Dear Hans-Peter Kubis, PD. Dr. rer. nat.

 Thank you for inviting us to submit a revised draft of our manuscript entitled, “Impact of acute complex exercise on inhibitory control and brain activation: a functional near-infrared spectroscopy study” to PLOS ONE (the revised title is “Impact of acute open-skill exercise on inhibitory control and brain activation: a functional near-infrared spectroscopy study”). We also appreciate the time and effort you and the reviewers have dedicated to providing insightful feedback on ways to strengthen our paper. Thus, it is with great pleasure that we resubmit our article for further consideration. And we sincerely apologize for being unable to resubmit the revision by the due day. 

We have incorporated changes that reflect the detailed suggestions you have provided. We also hope that our edits and the responses we provide below satisfactorily address all the issues and concerns you and the reviewers have noted. 

 To facilitate your review of our revisions, the following is a point-by-point response to the questions and comments delivered in your letter dated November 25, 2022.

Shinji Takahashi

Tohoku Gakuin University

2-1-1 Tenjinzawa, Izumi-ku, Sendai, Miyagi, 981-3193 Japan

Phone number: +81-22-773-3389

Email address: shinji@mail.tohoku-gakuin.ac.jp

 

Response to Reviewers

 We wish to express our appreciation to Reviewer 1 for their insightful comments on our paper. We feel the comments have helped us significantly improve the paper. And we sincerely apologize for being unable to resubmit the revision by the due date. Revision documents in the text are highlighted in “Revised Manuscript with Track Changes” in line with the guideline of PLOS ONE.

We changed the title and the measurements of the functional near-infrared spectroscopy (fNIRS) and most of the statistical models based on the reviewers’ suggestions. Although we reanalyzed the revised dataset and statistical models, the results for the significance of all independent variables confirmed our previous results. 

Reviewer 1 was also concerned about the quality of English writing. In response, we have carefully copy edited the manuscript.

Comment 1-1: 

L56. Complex term is likely not appropriate in the abstract and other places of the manuscript.

Motor skill execution is a fundamental part of sporting expertise and a number of recent studies have begun to examine the benefits of visuomotor skills, as those encountered in Badminton on cognitive function. Authors are encouraged to categorize differently exercises they studied.

Response 1-1

 In response to this comment, we exchanged the term “complex” exercise for “open-skill” exercise. In line with the comment, we exchanged “complex” for “open-skill” to improve readability. We also substituted “simple” exercise for “closed-skill” exercise. 

Comment 1-2: L62. Static is not the good term to use.

Response 1-2

 We chose the term “static” due to its use and definition in the previous work by Voss et al., 2010. This also applies to “strategic” and “interceptive.” Therefore, here we did not change the “static” term.

Comment 1-3: L64-72 and more (e.g. L 84-90, ..). Running cannot be considered simple, and badminton complex. Open skill versus closed skill sports should be used throughout the manuscript according to the scientific background in motor control. Please change accordingly.

Response 3

 As for “Response 1-1” above, we corrected “simple” to “closed-skill” and “complex” to “open-skill.”

Comment 1-4: Overall, introduction section is rather long. In the rationale, some parts could be removed; for instance, all sentences dealing with mental / cognitive demands and so mental fatigue are out of scope. Stay focus on the executive function of interest (inhibition) that can be modulated in badminton (not really described). Authors are presenting results of literature in executive function as a whole and with different physical tasks but they not addressed basically inhibition in open skill sports like Badminton or Tennis table.

Response 1-4

 In response to the first suggestion, we removed some sentences about mental/cognitive demands (e.g., P3) and non-essential descriptions (e.g., “on a motor-driven treadmill”). Throughout the introduction, we reworked to shorten the length by deleting some descriptions. Second, we did not delete the descriptions about cognitive demands and mental fatigue in the speculations by O’Leary et al. and Gallotta et al. because these descriptions relate to the possibility that a single bout of open-skill exercise could impact executive functions negatively. Third, to answer the reviewer’s point, we attempted to rework the explanation about the impacts of badminton on executive functions and the underpinning brain regions based on the sparse evidence in the literature. As we showed in the previous manuscript, the evidence that chronic effects of open-skill sports on executive functions and the associated brain regions are more substantial than closed-skill sports has been accumulated by cross-sectional studies. On the other hand, the evidence for any acute impacts of open-skill exercises is less well known and mixed. Pontifex et al. (2019) argued that it is inappropriate to cluster and compare studies assessing the acute impact of exercise and assessing the effects of regular exercise. Therefore, we did not present a more detailed hypothesis about the acute effects of open-skill exercises on inhibitory control and brain activation. We think our results expand knowledge about the relationship between the acute effects of open-skill exercises and cognition and the brain.

Comment 1-5: Brain activation is cited but what does it mean? Activation may be due to inhibition or facilitation…Some EEG studies are proposed while authors measured brain activity with other neuroimaging technique (fNIRS) that cannot give relevant information on the inhibitory processing (as the P3 component can do).

Response 1-5

 First, as per Response 1-4 above, we deleted descriptions of the EEG studies' results where possible. Second, we changed the term “brain activation” to “nervous activity” for electroencephalogram (EEG) or “hemodynamics” for fNIRS, depending on the measurement methods, throughout the revised manuscript. This rewording did not affect our consideration and discussion in this paper.

Comment 1-6: L130-132. At the end, hypothesis is not based on a strong rationale.

Response 1-6

 As Reviewer 1 pointed out, we did not formulate the research hypothesis with a strong rationale because the evidence for an effect of a single bout of open-skill exercise on cognitive functions and brain activation is insufficient at present. If the evidence for an effect of regular open-skill exercises can be extrapolated to the association between an acute open-skill exercise and inhibitory function and brain activation, we might formulate two contradictory hypotheses. On the one hand, a single bout of open-skill exercise could enhance inhibitory control performance with the same as the regular effects of open-skill exercise. On the other hand, even if a single bout of open-skill exercise also induces mental or cognitive fatigue, resulting in temporal deterioration of the inhibitory control performance, repeating the open-skill exercise for several months could benefit inhibitory control. However, given the insufficient evidence for acute impacts of open-skill exercise on executive functions, it is hard to formulate the research hypothesis with a strong rationale in this study.

 In our previous work, we have found that a single bout of badminton enhances inhibitory control (Takahashi and Grove, 2019 and 2020). Additional studies have reported that the enhancement of inhibitory control after a single bout of exercise is accompanied by increased hemodynamics in the prefrontal cortex. Given these results, our hypothesis that “an open-skill exercise like badminton would shorten reaction times in the Stroop task and increase hemodynamics in the prefrontal cortex compared to running” in the previous manuscript seems reasonable. Therefore, we did not change the description of the research hypothesis in the revised manuscript. 

Comment 1-7: Please rephrase / clarify and simplify the introduction section.

Response 1-7

 As for Reviewer 1’s suggestions above, we reworked the Introduction.

Comment 1-8: L143. how did authors assess ‘22’ as an adequate sample size? Please add details on the software used for conducting a Power analysis and the inputs variables (based on which references / findings? It is not reported).

Response 1-8

 We conducted the power analysis using the G*Power. We added the software name in the revision. Regarding the information needed for the power analysis, we reported the input variables and their references in the previous manuscript. We reported only partial eta squared as an effect size in the previous manuscript. We also added f values as an effect size directly used in the G*Power to make readers retest the power analysis quickly in the revised manuscript.

Comment 1-9: L190. How did authors match and control the workload (volume x intensity) between running and badminton? This is unclear. Table 2 indicates only intensities of each intervention. Ten minutes were used for all interventions?

Response 1-9

 As reported in the previous manuscript (L200-203 in the previous manuscript), we set the running speed according to each participant’s 75%VO2peak, which was previously shown to be an intensity equal to that of the badminton intervention (Takahashi and Grove, 2019, Takahashi and Grove, 2020). And we set the duration of all three interventions to ten minutes.

Comment 1-10: L204. Can you confirm that Oxygen uptake, carbon dioxide output and HR were monitored during the Badminton session? A portable device was used for that purpose. Please indicate the calibration phases for using the MetaMAx system.

Response 1-10

 We monitored VO2, VCO2, and HR during the Badminton intervention using the MetaMax 3B device. In accordance with the advice, we added the calibration procedures for the MetaMax 3B device.

Comment 1-11: L268-282. Overall, fNIRS processing does not appear to follow current recommendations.

Commnet 1-12: As recommended in the fNIRS field, HbO traces should be analysed concomitantly with HbR signals to make sure that neurovascular coupling response occurred; false positives are possible, especially by looking only at HbO.

Comment 1-13: In general, the data processing of the fNIRS data is well described. However, I miss a description of how the authors have dealt with systemic physiological artifacts that are known to confound the fNIRS signal (Scholkmann, Tachtsidis, Wolf, & Wolf, 2022; Tachtsidis & Scholkmann, 2016; Yücel et al., 2021).

Comment 1-14: Can you confirm that the authors use a lowpass filter around 0.09 Hz to account for the Mayer-Waves as recommended in the literature? (Pinti, Scholkmann, Hamilton, Burgess, & Tachtsidis, 2019).

Response 1-11 – 1-14

 Thank you for your insightful suggestions. We will respond to Comments 1-11 to 14 together. First, regarding the setting of the band-pass filter, we changed the thresholds of high-pass and low-pass filters from “0.005 Hz and 0.05 Hz” to “0.01 Hz and 0.1 Hz” based on the previous review paper (Herold et al., 2018). We also added Herold et al. (2018) to the reference list.

 Second, regarding how to deal with systemic physiological artifacts, we rewrote the explanation of the “hemodynamic modality separation method” (Yamada et al., 2012). In the previous manuscript, we just wrote, “artifacts from changes in skin blood flow in the forehead were removed using the hemodynamic modality separation method.” The systemic physiological artifacts evoked by a body’s motion and psychophysiological changes show up in skin blood flow changes. In short, the hemodynamic modality separation method is the method to deal with systemic physiological artifacts. In order to clarify that the hemodynamic modality separation method removes systemic physiological artifacts, we rewrote it as “Moreover, systemic physiological artifacts were removed using the hemodynamic modality separation method [40].”

 The hemodynamic modality separation method decomposes hemoglobin measures into systemic and functional components. While oxy-Hb kinetics positively correlates to deoxy-Hb kinetics in the systemic component, oxy-Hb kinetics is negatively coupled with deoxy-Hb kinetics in the functional component. Therefore, as we reported in the previous manuscript, after transforming measures to z-scale, functional oxy-Hb and deoxy-Hb in the interest of this study were completely equivalent values, but those signs were inverse. The data processing above can reduce the false positive rate.

Comment 1-15: L296. Please add all information (checking) regarding the use of parametric test (one-way repeated ANOVA for physiological variables; two-way repeated ANOVA for psychological moods scores, three-way for the Stroop tasks, fNIRS variables etc.): normality issue, homogeneity of variance. Effect size (partial eta squared) for the main factors should be indicated thereafter in the results.

Response 1-15

 First, we changed the independent variable and statistical analysis to compare the accuracy of the Stroop task performance between interventions. We employed “number of errors” instead of “accuracy rate (%).” We then changed the analysis model from the mixed model to the log-linear mixed model, which is an adequate model for frequency data. The results of the log-linear mixed mode were the same as the previous results of the accuracy rate by the mixed model.

 On the other hand, we did not add information about the normality of dependent variables, homogeneity of variance, and effect sizes for the mixed models. For the normality of dependent variables, we checked the distribution of each dependent variable but did not report them in the previous manuscript for the following reasons. Because the dependent variables were repeated measures, the normality tests should be conducted by each mode for intensity variables, by each mode and time for mood, each mode, and by time and task condition for the reaction time and hemodynamics. Finally, the number of results for normality tests was 72. As a result, nineteen of 72 variables (26.4%) did not follow the normality assumption. To cope with the violation, we modified the variables using logarithm transformation or arcsine transformation when even one in a series of repeated measure variables did not follow the normal distribution. We then analyzed using the transformed variables, but the results for the significance of the main effects and interactions were the same as before the transformation. Therefore, we did not report the results of the normality test. 

 We think reporting the homogeneity of variance (maybe sphericity assumption here) and the effect sizes seem inappropriate. Our study analyzed the ANOVA models using the mixed model with the covariance structure as unstructured. Unstructured allows us to use ANOVA models even when the sphericity assumption, which differences between all pairs in repeated measures are identical, is violated. Furthermore, calculating an effect size for the mixed model is complicated and not prevalent. Well-known effect sizes (e.g., Cohen’s d, eta squared) for linear models (e.g., t-test, ANOVA) assume that there are one or more fixed effects and only one random effect (error variance). The effect sizes are obtained as a ratio of the fixed effect and the sum of the fixed effect and error variance. However, unlike linear models, the mixed model includes one or more random effects. Because of multiple random effects, we cannot simply calculate the effect sizes. Instead of the effect size for the main effects and interactions, we reported Cohen’s ds to present how large differences between pairs with significance. 

Comment 1-16. Did you test some interaction factors according to your hypothesis. Please elaborate.

Response 1-16

 We hypothesized that the badminton intervention makes a difference in the Stroop task performance and hemodynamics from running and seated rest interventions, then we addressed the significant interactions regarding Mode (Badminton, Running, and Control). We reported the analyses and those results in the previous manuscript (L317-320; L355-362; L375-383 in the previous manuscript). 

Comment 1-17. Individual data should be presented in figures.

Response 1-17

 In response to this comment, we tried to rework the graphs by adding individual plots. Below is a graph of changes in reaction time by adding individual data.

The reworked graph might be informative; however, the graph does not seem to be suitable for presenting the interaction of Mode and Time, which we would like to show the most. Therefore, we did not add individual data to the graphs in the revised manuscript.

Comment 1-18. L447. Intensity (Table 2: 77-78% peak VO2, 84-86% HR, RER close to 0.97 and RPE around 13-14) is claimed vigorous by authors. It is quite surprising to observe a decoupling between RPE values and physiological variables like HR. Please comment.

Response 1-18

 We think this study's correspondence between HR and RPE is standard. The coupling of HR and RPE, in which one-tenth of HR value is close to RPE, appears in a graded exercise test or cycling. However, it is known that one-tenth of HR values are higher than RPE during running with a constant load. For example, Larsen et al. (2002) studied HR and RPE for 104 young adults, reporting that when the participants jogged for 14.1 min (N = 56), while HR elevated to 180.9 bpm (93.1% HRmax), RPE was 13.2. Even at walking for 19.9 min (N = 17), HR and RPE were 162.4 bpm (81.5% HRmax) and 12.1, respectively. On the other hand, at the end of the graded exercise test, HRmax was 195.3 bpm with RPE max 19.3. The coupling of HR and RPE during exercises differs from exercise modes or protocols. Therefore, we did not add comments for the coupling between HR and RPE in the revised manuscript.

Larsen, G. E., George, J. D., Alexander, J. L., Fellingham, G. W., Aldana, S. G., & Parcell, A. C. (2002). Prediction of maximum oxygen consumption from walking, jogging, or running. Research quarterly for exercise and sport, 73(1), 66-72.

Comment 1-19. Can you propose dedicated reference for this statement on vigorous? Ref 41 is not up to date.

According to Jones and Poole, domains of exercise are low/moderate/heavy and severe.

Response 1-19

 First, we apologize for the typo on Ref.41 in the previous manuscript. “Medicine ACoS” as the authors’ name was wrong, but the correct one is “American College of Sports Medicine.” 

The domains of exercise intensity Reviewer 1 suggested are based on oxygen uptake kinetics or steady state for blood lactate. Poole and Jones (2012) defined “moderate” below the lactate threshold (LT) or gas exchange threshold (GET). Similarly, “heavy” is defined as more than the LT or GET and less than the maximal lactate steady state (critical power), and “severe” is over the critical power. The intensity category by Jones and Poole is frequently used to study VO2 kinetics, especially analyzing the slow component of VO2, whose time constant and amplitude indicate the types and tolerance of muscles. The category by Jones and Poole does not seem to match the goal of this study.

The category of exercise intensity by ACSM’s guidelines based on HRmax and VO2max is frequently referred to in the cognition-exercise field. Reporting exercise intensities based on the ACSM’s guidelines would allow readers to intuit the intensity of each intervention in this study is. Therefore, we did not change the explanation about interventions’ intensities.

Poole, D. C., & Jones, A. M. (2012). Oxygen uptake kinetics. Compr Physiol, 2(2), 933-996.

Comment 1-20. L461-463. Please develop these first ideas for the main significant result you observed.

Response 1-20

 A new finding of this study was that improved inhibitory control after an acute open-skill exercise does not associate with hemodynamics in the prefrontal cortex. After reporting the results for fNIRS, we reworked discussions about the association between acute open-skill exercise and executive function by adding citations to strengthen the rationale. 

Comment 1-21. L472. Authors did not propose some hypothesis on such a correlation; no correlation results were reported.

Response 1-21

 We thank you for the point. To answer this point, we corrected this sentence by changing the term “correlate” to “coincide.”

Comment 1-22. L498. Since blood lactate samples were not measured, authors cannot state that. Please remove this speculation, out of scope.

Response 1-22

 As Reviewer 1 pointed out, we did not measure the blood lactate. Nevertheless, in the previous manuscript, we stated that “serum lactate concentration during badminton was comparable with during running.” in L498. Therefore, we revised the sentence to, “serum lactate concentration during badminton seemed to be comparable with during running.”

On the other hand, we think we can mention the lactate hypothesis proposed by Kao et al. Carbon dioxide output during exercises is determined by the metabolic and non-metabolic CO2 output. The metabolic CO2 is produced by oxidation to yield energy. When the oxidation substrate is glucose, metabolic CO2 equals VO2, while when the oxidation substrate is a fatty acid, that is 0.7 times VO2. In short, the value of metabolic VCO2 varies equally or slightly less than VO2. On the other hand, non-metabolic VCO2 is produced by the bicarbonate buffer system. To remove the proton that the lactate releases, the bicarbonate binds the proton to balance the pH in the blood, resulting in non-metabolic CO2. Due to the non-metabolic CO2, VCO2 during exercise can be higher than VO2, and the respiratory exchange ratio (RER) can be more than 1.0. In other words, RER during exercise is not over 1.0 when the lactate is not increased, excluding the transient phases such as the beginning and just after exercise cessation. The measures of RER we obtained for the badminton and running interventions were 0.96 and 0.98, respectively. Both were less than 1.0 and comparable. Considering the features of RER above, it would be reasonable to mention the lactate hypothesis based on RER values. Therefore, we did not remove this part in the revised manuscript.

Comment 1-23. From L 500. This section is quite difficult to follow. Please simplify it. When authors are dealing with brain activation, please be accurate on the variables/methods/paradigms used.

Response 1-23

 Thank you for this comment. We have revised this paragraph. For example, we used the term “brain activation” in the introduction and discussion by not distinguishing the nervous activity measured by electroencephalography (EEG) and hemodynamics by fNIRS. To correct this ambiguity, we changed “brain activation” to “nervous activity” for EEG measures and “hemodynamics” for fNIRS measures as possible. And we added that the previous study indicated a spatial correlation between nervous activity by EEG and hemodynamics by fNIRS in order to strengthen the rationale of the discussion in this paragraph. Furthermore, we removed some sentences to simplify the discussion.

Comment 1-24. There are other limitations related to fNIRS processing (see before). Event-related design with fNIRS is not appropriate as compared to EEG for unveiling underlying mechanisms on the brain changes related to the inhibitory control.

Response 1-24

 We agree with this point. fNIRS is inferior to the EEG in temporal resolution. Event-related potentials like P3 are registered on a milli-second scale while detecting changes in hemodynamics for the fNIRS require 10 second or more. Although the difference in temporal resolution between EEG and fNIRS is obvious, studies in which EEG and fNIRS were simultaneously measured in event-related design experiments found significant correlations between the P3 component by EEG and changes in hemodynamics by fNIRS (Lin, Sai, & Yuan, 2018; Rizer, Aday, & Carlson, 2018). Given the previous results, although we should pay attention to the difference in temporal resolutions for EEG and fNIRS, employing the NIRS to investigate the brain function related to inhibitory control would not be a limitation for this study.

Lin, X., Sai, L., & Yuan, Z. (2018). Detecting concealed information with fused electroencephalography and functional near-infrared spectroscopy. Neuroscience, 386, 284-294. 

Rizer, W., Aday, J. S., & Carlson, J. M. (2018). Changes in prefrontal cortex near infrared spectroscopy activity as a function of difficulty in a visual P300 paradigm. Journal of Near Infrared Spectroscopy, 26(4), 222-228. 

Comment 1-25. Reference format: insert a space before the bracket for reference lines 44, 48, 50 and so on.

Response 1-25

 Thank you for the detailed and careful points. In accordance with the suggestion, we corrected them.

Comment 1-26. L195 simple without s

Response 1-26

We think the reviewer is referring to the word “singles” which refers to a one on one badminton match. We have revised the sentence to clarify”

In the badminton intervention, the participants played one singles game against an…

Comment 2-1. The sentence starting with “Despite contrary evidence” line 65 page 4 was very confusing and seemed out of place. It’s not clear whether the authors of this paper disagree with Diamond and Ling’s conclusions, or if they are referencing the controversies mentioned in Diamond and Lin’s paper.

Response 2-1

 Thanks for this comment. This sentence was deleted as a result of the extensive revisions we made to the manuscript.

Comment 2-2. Figure 2: the circle that is meant to show the head seems to be missing some landmarks like a nose, ears, etc. As it is now it is not very informative and it is hard to understand how low on the forehead the optodes were placed. There also appear to be typos in the legend, should be Emitter Probe and Detector Probe (or just Emitter and Detector).

Response 2-2

 We thank you for the careful advice and are sorry for the typos in Figure 2. In accordance with the advice, we added a nose and ears in Figure 2. We also corrected the typos by changing “prove” to “probe.”

Comment 2-3. Was it possible to determine if participant’s heart rates had returned to their baseline before the Stroop task was started in the post intervention run?

Response 2-3

 In response to this comment, we added the potential of elevated HR after exercise interventions in the limitations.

Comment 2-4. The top bandpass parameter, 0.05 Hz, is quite low. To avoid respiration artifact being below 0.3 Hz should be sufficient. Given the bandpass parameters it would be good to be clear about how long the condition blocks are. It appears that they are at least 3 sec x 24 trials but it’s unclear how long the response period is. Was there only one block of each condition? For future studies you may consider a more classic “block design” with multiple blocks of 16-30 sec in duration.

Response 2-4

 We thank you for your specific advice. We changed the thresholds of high-pass and low-pass filters from “0.005 Hz and 0.05 Hz” to “0.01 Hz and 0.1 Hz” based on the previous review paper (Herold et al., 2018). We also added Herold et al. (2018) to the reference lists. Regarding the response period for the Stroop task, in accordance with the suggestion, we added the response period was 2.5 s in the “Color-word Stroop task” section.

 Regarding the number of blocks composed in the Stroop task, we employed only one block for each neutral and incongruent block in this study. It is a shame we have already completed more than half of the experiments for the following research using the same Stroop task in this study. After the following research, we will employ the more classic block design with multiple blocks.

Comment 2-5. Were any motion artifacts observed in the fNIRS data, and if so how were they handled?

Response 2-5

In response to this comment, we rewrote the explanation of the “hemodynamic modality separation method” (Yamada et al., 2012). The new text reads:

“Moreover, systemic physiological artifacts were removed using the hemodynamic modality separation method [40].”

Comment 2-6. There are many explanatory variables in the model (Condition, time, mode, interactions, order of interventions, order of condition) for a comparatively small number of participants (24). The authors should justify that their data supports this complex of a model or consider removing variables.

Response 2-6

 In accordance with this suggestion, we removed the orders of interventions and conditions as independent variables. Removing these variables from the model did not affect the results of the significance for all main effects and interactions. We reported the reworked results in the revised manuscript. We also reported that there was no influence of the orders of interventions and conditions on the results in the first of the Results section.

Comment 2-7. Did the group performance (accuracy, reaction time) on the Stroop task match what was expected based on the literature?

Response 2-7

 Thank you for your asking. Stroop task performance was consistent with the literature in that the reaction time and accuracy for the incongruent condition were significantly longer and lower than for the neutral condition. On the other hand, the difference between the neutral and incongruent conditions in this study seems small compared with previous results (Byun et al., 2014; Schroeter, Zysset, Wahl, & von Cramon, 2004). The overall difference in reaction time between the neutral and incongruent conditions in this study was Cohen’s d = 0.62, while the previous results in Byun et al. and Schroeter et al. reported Cohen’s ds were more than 0.80. The gap between the previous studies and this study shows the potential that the Stroop task in this study is easier than the literature. As the possibility could be a part of the limitations, we added the possibility in the limitation section. We also added Schroeter et al (2004) to the reference list.

Schroeter, M. L., Zysset, S., Wahl, M., & von Cramon, D. Y. (2004). Prefrontal activation due to Stroop interference increases during development—an event-related fNIRS study. NeuroImage, 23(4), 1317-1325.

Comment 2-8. Did the difference in brain activation between Stroop conditions for left prefrontal cortex only match existing literature?

Response 2-8

 Thank you for your asking again. The result that the left prefrontal cortex was more activated during the incongruent condition than during the neutral condition matches the literature, demonstrating the reliability of the fNIRS measure in this study. Therefore, we added the description that the left prefrontal cortex was activated during the incongruent condition related to the neutral condition to the discussion of fNIRS measures, suggesting the reliability of the fNIRS measures in this study.

Comment 2-9. For Figure 4, I’d suggest using a different color scheme or even grayscale as blue and red are traditionally used to represent oxy- and deoxyhemoglobin. You could also consider putting the Left prefrontal cortex graphs on the left of the figure, and the right prefrontal graphs on the right.

Response 2-9

 Thank you for your helpful advice. In accordance with the advice, we reworked Figure 4 using grayscale and changed the arrangement of each graph.

Thank you again for your comments on our paper. We trust that the revised manuscript is suitable for publication.

---

## [Decision Letter · Decision Letter 1]

14 Mar 2023

Impact of acute open-skill exercise on inhibitory control and brain activation: a functional near-infrared spectroscopy study

PONE-D-22-26975R1

Dear Dr.Takahashi,

We’re pleased to inform you that your manuscript has been judged scientifically suitable for publication and will be formally accepted for publication once it meets all outstanding technical requirements.

Kind regards,

Hans-Peter Kubis, PD. Dr. rer. nat.

Academic Editor

PLOS ONE

Additional Editor Comments (optional):

Reviewers' comments:

Reviewer's Responses to Questions

**Comments to the Author**

1. If the authors have adequately addressed your comments raised in a previous round of review and you feel that this manuscript is now acceptable for publication, you may indicate that here to bypass the “Comments to the Author” section, enter your conflict of interest statement in the “Confidential to Editor” section, and submit your "Accept" recommendation.

Reviewer #1: All comments have been addressed

Reviewer #2: All comments have been addressed

2. Is the manuscript technically sound, and do the data support the conclusions?

Reviewer #1: Yes

Reviewer #2: Yes

3. Has the statistical analysis been performed appropriately and rigorously? 

Reviewer #1: Yes

Reviewer #2: Yes

4. Have the authors made all data underlying the findings in their manuscript fully available?

Reviewer #1: Yes

Reviewer #2: Yes

5. Is the manuscript presented in an intelligible fashion and written in standard English?

Reviewer #1: Yes

Reviewer #2: Yes

6. Review Comments to the Author

Reviewer #1: (No Response)

Reviewer #2: The authors have substantially revised and improved the manuscript according to reviewer comments and therefore I recommend that the manuscript should be accepted.

7. PLOS authors have the option to publish the peer review history of their article (what does this mean?). If published, this will include your full peer review and any attached files.

Reviewer #1: No

Reviewer #2: No

---

## [Editor Report · Acceptance letter]

16 Mar 2023

PONE-D-22-26975R1 

Impact of acute open-skill exercise on inhibitory control and brain activation: a functional near-infrared spectroscopy study 

Dear Dr. Takahashi:

I'm pleased to inform you that your manuscript has been deemed suitable for publication in PLOS ONE. Congratulations! Your manuscript is now with our production department. 

Kind regards, 

on behalf of

Dr. Hans-Peter Kubis 

Academic Editor

PLOS ONE